# An Updated Systematic Review and Network Meta-Analysis of First-Line Triplet vs. Doublet Therapies for Metastatic Hormone-Sensitive Prostate Cancer

**DOI:** 10.3390/cancers17020205

**Published:** 2025-01-09

**Authors:** Akihiro Matsukawa, Giulio Litterio, Angelo Cormio, Marcin Miszczyk, Mehdi Kardoust Parizi, Tamás Fazekas, Ichiro Tsuboi, Stefano Mancon, Robert J. Schulz, Ekaterina Laukhtina, Paweł Rajwa, Keiichiro Mori, Piotr Chlosta, Michele Marchioni, Luigi Schips, Jun Miki, Takahiro Kimura, Shahrokh F. Shariat, Takafumi Yanagisawa

**Affiliations:** 1Department of Urology, Comprehensive Cancer Center, Medical University of Vienna, 1090 Vienna, Austria; akihiro.matsukawa@meduniwien.ac.at (A.M.); giulio.litterio@gmail.com (G.L.); angelo.cormio@meduniwien.ac.at (A.C.);; 2Department of Urology, The Jikei University School of Medicine, Tokyo 105-8461, Japantkimura@jikei.ac.jp (T.K.); 3Department of Medical, Oral and Biotechnological Sciences, G. d’Annunzio University of Chieti, 66013 Chieti, Italy; 4Department of Urology, Azienda Ospedaliero-Universitaria Ospedali Riuniti Di Ancona, Università Politecnica Delle Marche, 60126 Ancona, Italy; 5Collegium Medicum—Faculty of Medicine, WSB University, 41-300 Dąbrowa Górnicza, Poland; 6Department of Urology, Semmelweis University, 1082 Budapest, Hungary; 7Department of Urology, Shimane University Faculty of Medicine, Izumo 693-8504, Shimane, Japan; 8Department of Biomedical Sciences, Humanitas University, 20072 Pieve Emanuele, Italy; 9Department of Urology, IRCCS Humanitas Research Hospital, 20090 Milan, Italy; 10Department of Urology, University Medical Center Hamburg-Eppendorf, 20251 Hamburg, Germany; 11Second Department of Urology, Centre of Postgraduate Medical Education, 01-813 Warsaw, Poland; 12Division of Surgery and Interventional Science, University College London, London WC1E 6BT, UK; 13Department of Urology, Medical College, Jagiellonian University, 30-688 Krakow, Poland; 14Department of Urology, The University of Texas Southwestern Medical Center, Dallas, TX 75390, USA; 15Department of Urology, Weill Cornell Medical College, New York, NY 10065, USA; 16Department of Urology, Second Faculty of Medicine, Charles University, 15006 Prague, Czech Republic; 17Division of Urology, Department of Special Surgery, The University of Jordan, Amman 11942, Jordan; 18Karl Landsteiner Institute of Urology and Andrology, 1010 Vienna, Austria; 19Research Center for Evidence Medicine, Urology Department, Tabriz University of Medical Sciences, Tabriz 51656-65811, Iran

**Keywords:** mHSPC, ARPI, docetaxel, triplet therapy, progression-free survival, overall survival, adverse event, systematic review, network meta-analysis

## Abstract

We conducted an updated systematic review and network meta-analysis, including the recently published ARANOTE trial, to evaluated the efficacy and safety of triplet therapy for metastatic hormone-sensitive prostate cancer. Triplet therapies demonstrated significant improvements in PFS compared to ARPI-based doublet therapies, particularly in patients with high-volume disease. Conversely, their benefit in low-volume disease remains unclear. No significant difference was observed in the incidence of severe adverse events between triplet and ARPI-based doublet therapies. Further head-to-head trials are needed to provide more robust evidence.

## 1. Introduction

The development of androgen receptor pathway inhibitors (ARPIs), such as abiraterone (Abi), apalutamide (Apa), darolutamide (Dar), and enzalutamide (Enz), revolutionized the treatment for patients with metastatic, hormone-sensitive prostate cancer (mHSPC). Current clinical guidelines regard ARPI-based treatments as the standard of care (SOC), including doublet therapies (ARPI + ADT) and triplet therapies (ARPI + docetaxel [Doc] + androgen deprivation therapy [ADT]), considered for eligible patients [1,2,3]. However, it is not entirely clear which regimens are superior in which clinical situations.

Our previous network meta-analysis (NMA) indicated that triplet therapy may be preferable for patients with high-volume mHSPC, while ARPI-based doublet therapy is recommended in patients with low-volume disease, due to the uncertain benefit of the triplet-therapy approach in these cases [4,5]. The recent publication of the ARANOTE trial [6], the first to evaluate Dar + ADT in mHSPC, provided novel insight in this context. To further clarify the comparative efficacy and safety of available combination therapies, we conducted an updated NMA that included the ARANOTE trial [6].

## 2. Materials and Methods

The study protocol is registered with the International Prospective Register of Systemic Reviews database (PROSPERO: CRD42024591458). This meta-analysis follows the guidelines of the Preferred Reporting Items for Systematic Reviews and Meta-Analyses (PRISMA) and adheres to the AMSTAR2 checklist [7,8] (Appendix A).

### 2.1. Study Selection and Characteristics

On 17 September 2024, we conducted comprehensive searches in MEDLINE (via PubMed), Embase, and the Web of Science Core Collection to identify studies on systemic therapy for mHSPC. The complete search strategy is provided in Appendix A. Two investigators independently screened titles and abstracts for eligibility, with discrepancies resolved by consensus among the co-authors.

### 2.2. Inclusion and Exclusion Criteria

The PICOS framework was applied to address our clinical questions [9]. We included studies investigating patients with mHSPC (population) treated with systemic therapies (interventions) in comparison to alternative systemic therapies (comparisons). The primary endpoint was progression-free survival (PFS), with the secondary endpoints including overall survival (OS) and adverse event (AE) incidence (outcomes), as reported in the phase Ⅲ randomized controlled trials (RCTs) (study). The exclusion criteria comprised reviews, editorial comments, case reports, books, replies to authors, meeting abstracts, and articles not published in English. Additionally, studies evaluating agents not approved by the Food and Drug Administration (FDA) or European Medicines Agency (EMA) were excluded.

### 2.3. Data Extraction

Two authors independently extracted the data, including the first author’s name, publication year, recruitment period, study design, inclusion criteria, number of patients, patient’s age, oncological outcomes (hazard ratios [HRs] with 95% confidence intervals [CIs]), and AE incidence. The AE classification and grades followed the National Cancer Institute Common Terminology Criteria for Adverse Events (NCI-CTCAE). In instances where the cohort data overlapped, the most recent and appropriate articles were selected. In studies that prior or concomitant Doc use was permitted, we extracted data from a subgroup analysis stratified by the presence or absence of Doc. If the subgroup analyses stratified by disease volume did not provide separate data for patients with and without Doc use, those analyses were excluded from the volume-based subgroup analysis. Discrepancies were resolved through discussion.

### 2.4. Risk of Bias Assessment

Two authors independently evaluated each study, utilizing the Risk-of-Bias version 2 (ROB2) tool [10] for RCTs (Appendix A). Conflicts were resolved by mediation with co-authors.

### 2.5. Network Meta-Analysis

We conducted an NMA with frequentist random-effects models to enable both direct and indirect comparisons between treatments [11,12]. A network diagram illustrated the treatment connections (Appendix A). The oncological outcomes were evaluated through contrast-based analyses using log-transformed HR and corresponding standard errors calculated based on published HR and their 95% CIs [13]. To assess AE incidence, arm-based analyses were performed, estimating risk ratios (RRs) and 95% CIs based on data from the included articles. The relative effects were reported as HRs or RRs with 95% CIs, with a statistically significant set at *p* < 0.05. A forest plot was created to visually represent the treatment effects. We assessed heterogeneity among studies using Cochran’s Q test, with *p* < 0.05 indicating significant heterogeneity. Treatment rankings for each outcome were determined based on the P-score [14]. Subgroup analyses were conducted for patients with high- versus low-volume disease, as defined by the CHARRTED trial criteria. The high-volume disease was characterized by the presence of visceral metastases, or four or more bone metastases, of which, at least one must be located outside the vertebral column or pelvic bone [15]. All the statistical analyses were performed using R version 4.3.0 (R Foundation for Statistical Computing, Vienna, Austria) with the “netmeta” package.

## 3. Evidence Synthesis

### 3.1. Study Selection and Characteristics

The search strategy is outlined in Figure 1. A total of 12 RCTs [6,16,17,18,19,20,21,22,23,24,25,26], comprising 11,954 patients with mHSPC, were included based on our inclusion and exclusion criteria. The ARASENS [16], ARCHES [17], ENZAMET [19], PEACE-1 [22], STAMPEDE arm J [27], and TITAN [26] trials permitted the prior or concomitant administration of Doc. As the STAMPEDE arm J [27] did not provide a subgroup analysis by Doc status, it was excluded from our analysis. Additionally, the ARCHES [17] and TITAN [26] trials did not stratify by both disease volume and Doc status, leading to their exclusion from our subgroup analysis. To prevent cohort duplication, we specifically selected STAMPEDE arm CG [24] for comparison. However, since stratified data by disease volume were unavailable in this trial, we included STAMPEDE arm G and BCE in the subgroup analysis [23,28]. The ARASENS trial [16] did not provide data on commonly defined PFS, thereby preventing the inclusion of the Dar + Doc + ADT in the PFS analysis. Detailed patient characteristics and outcomes are provided in Table 1.

### 3.2. Assessment of Risk of Bias and Quality of Study

The risk-of-bias assessments for each study are summarized in Appendix A. The RoB2 tool classified four studies as having some concerns and eight studies as having a low risk of bias.

### 3.3. Network Meta-Analysis

#### 3.3.1. Progression-Free Survival

##### Overall Patients

A total of nine trials [6,17,18,19,20,21,22,24,26], comprising 7646 patients with mHSPC, were included in our analysis to assess PFS. Our NMA revealed that triplet therapy significantly improved PFS compared to both Doc-based doublet therapy (HR: 0.51; 95% CI: 0.43–0.60; *p* < 0.001) and ARPI-based doublet therapy (HR: 0.74; 95% CI: 0.59–0.93; *p* = 0.01) (Figure 2). In addition, Figure 3 presents the results of individual regimen comparisons, showing that each evaluated regimen improved PFS compared to ADT alone. Conversely, there was insufficient evidence to show that any of the triplet therapies significantly improved PFS compared to Enz + ADT (Abi + Doc + ADT, HR: 0.85; 95% CI: 0.55–1.30. Apa + Doc + ADT, HR: 0.80; 95% CI: 0.36–1.76. Enz + Doc + ADT, HR: 0.87; 95% CI: 0.64–1.17) (Table 2). Based on the P-score analysis for treatment rankings in terms of PFS, triplet therapy had a high likelihood of providing the maximal PFS benefit: Enz + Doc + ADT (0.833), Abi + Doc + ADT (0.829), and Apa + Doc + ADT (0.80) (Table 3). NMAs focusing on radiographic progression-free survival (rPFS) are presented in Appendix A. There was no evidence of significant heterogeneity in any of the analyses.

##### Patients with High-Volume Disease

Among the patients with high-volume mHSPC (n = 7881), triplet therapy was associated with significant improvements in PFS compared to both Doc-based doublet therapy (HR: 0.50; 95% CI: 0.39–0.64; *p* < 0.001) and ARPI-based doublet therapy (HR: 0.64; 95% CI: 0.47–0.88; *p* < 0.01) (Figure 2). For individual regimens, there was insufficient evidence to confirm that adding Doc to any particular ARPI + ADT regimen results in a statistically significant improvement in PFS compared to Enz + ADT, including Abi + Doc + ADT (HR: 0.75; 95% CI: 0.42–1.34) and Enz + Doc + ADT (HR: 0.81; 95% CI: 0.50–1.32) (Table 2). In the treatment ranking based on the P-score analysis, Abi + Doc + ADT had the highest likelihood of improving PFS (0.91), followed by Enz + Doc + ADT (0.86) (Table 3). There was no evidence of significant heterogeneity.

##### Patients with Low-Volume Disease

In the patients with low-volume mHSPC (n = 7881), triplet therapy was associated with a significant improvement in PFS compared to Doc-based doublet therapy (HR: 0.45; 95% CI: 0.27–0.76; *p* < 0.01); however, no significant difference was observed between triplet therapy and ARPI-based doublet therapy (HR: 0.86; 95% CI: 0.45–1.67; *p* = 0.7). According to our NMA comparing individual regimens, neither Abi + Doc + ADT (HR: 1.55; 95% CI: 0.63–3.81) nor Enz + Doc + ADT (HR: 0.99; 95% CI: 0.43–2.28) showed significant improvements in PFS compared to Enz + ADT (Table 2). Our treatment ranking, based on the P-score analysis, showed that Enz + ADT had the highest likelihood of improving PFS (0.81), followed by Enz + Doc + ADT (0.80) (Table 3). There was no evidence of significant heterogeneity in any of the analyses.

#### 3.3.2. Overall Survival

##### Overall Patients

A total of ten studies [6,17,18,19,20,21,22,23,25,26], comprising 8951 patients, were included in our NMA assessing OS. As shown in Figure 2, triplet therapy reduced the risk of death by approximately 18% compared to ARPI-based doublet therapy, but the difference did not reach a conventional level of statistical significance (HR: 0.82; 95% CI: 0.67–1.01; *p* = 0.059). Triplet therapy was associated with a statistically significant improvement in OS compared to Doc-based doublet therapy (HR: 0.74; 95% CI: 0.66–0.83; *p* < 0.01). Based on the subgroup NMA of individual regimens, we did not find evidence of significant improvements for Abi + Doc + ADT (HR: 0.90; 95% CI: 0.65–1.25), Dar + Doc + ADT (HR: 0.81; 95% CI: 0.61–1.08), or Enz + Doc + ADT (HR: 0.96; 95% CI: 0.69–1.32) (Table 2). The P-score-based treatment ranking showed that Dar + Doc + ADT had the highest likelihood of providing the maximal OS benefit (0.93), followed by Abi + Doc + ADT (0.79) (Table 2). There was no evidence of significant heterogeneity in any of the analyses.

##### Patients with High-Volume Disease

For the patients with high-volume disease (n = 9504), triplet therapy did not show a statistically significant improvement compared to ARPI-based doublet therapy (HR: 0.82; 95% CI: 0.65–1.04; *p* = 0.1), while a significant improvement was observed compared to Doc-based doublet therapy (HR: 0.73; 95% CI: 0.64–0.84; *p* < 0.001) (Figure 2). Additionally, no statistically significant improvements in terms of OS were observed for Abi + Doc + ADT (HR: 0.76; 95% CI: 0.48–1.20), Dar + Doc + ADT (HR: 0.73; 95% CI: 0.48–1.10), or Enz + Doc + ADT (HR: 0.91; 95% CI: 0.57–1.46) (Table 2). Based on the P-score-based treatment ranking, Dar + Doc + ADT (0.91) had the highest likelihood of improving OS, followed by Abi + Doc + ADT (0.84) (Table 3). We did not find evidence for significant heterogeneity.

##### Patients with Low-Volume Disease

Among the patients with low-volume disease (n = 9504), triplet therapy was associated with statistically significant improvements in terms of OS compared to Doc + ADT (HR: 0.71; 95% CI: 0.52–0.97; *p* = 0.03), while no statistically significant improvement was observed compared to ARPI-based doublet therapy (HR: 1.03; 95% CI: 0.66–1.61; *p* = 0.9) (Figure 2). Based on the NMA of individual regimens, none of the triplet regimens showed statistically significant improvements in OS compared to Enz + ADT (Abi + Doc + ADT HR: 1.48; 95% CI: 0.75–2.90; Dar + Doc + ADT HR: 1.21; 95% CI: 0.62–2.37; Enz + Doc HR: 1.09; 95% CI: 0.52–2.29) (Table 2). The P-score treatment ranking revealed that Enz + ADT had the highest likelihood of providing the maximal OS benefit (0.85), followed by Enz + Doc + ADT (0.75) (Table 3). We did not find evidence for significant heterogeneity.

#### 3.3.3. Adverse Events

The available results from eight studies were included in these NMAs. As shown in Figure 2, no statistically significant differences were found in the incidence of either any-grade or grade ≥ 3 AEs when comparing triplet therapy, ARPI-based doublet therapy, and Doc-based doublet therapy. Based on the P-score-based ranking, Dar + ADT (0.63) had the highest likelihood of being associated with the lowest incidence of grade ≥ 3 AEs among the combination therapies, followed by Apa + ADT (0.62). The other relevant AE profiles are summarized in Appendix A.

## 4. Discussion

This updated NMA evaluates the efficacy and safety of triplet therapies compared to doublet therapies as first-line treatments for patients with mHSPC. Our study yielded several key findings. First, triplet therapy was associated with a significant improvement in PFS compared to both Doc-based and ARPI-based doublet therapy. Additionally, triplet therapy significantly improved OS compared to Doc-based doublet therapy. While it did not reach conventional statistical significance compared to ARPI-based doublet therapy, triplet therapy still demonstrated a beneficial effect. Second, in the patients with high-volume disease, triplet therapy improved PFS compared to both Doc-based and ARPI-based doublet therapy, while no statistically significant improvement was observed compared to ARPI-based therapy among patients with low-volume disease. Third, Dar + ADT exhibited the most favorable safety profile in grade ≥ 3 AE among combination therapies.

Our NMA revealed that triplet therapy significantly reduced the risk of progression or death by 49% and the risk of death by 26%, compared to Doc-based doublet therapy. Compared to ARPI-based doublet therapy, triplet therapy significantly reduced the risk of progression or death by 26% and the risk of death by 18%, although not reaching the conventional levels of statistical significance for the latter. OS is a clearly defined and clinically meaningful endpoint, but it has the disadvantages of requiring a large trial size and an extended follow-up period, leading to long study durations in mHSPC trials. Based on findings by Halabi et al. [29], PFS is a valid surrogate for OS in mHSPC patients and can be used to expedite drug development. While some concerns remain regarding the quality of the data on which this surrogacy was established [30], we decided to use PFS as the primary endpoint of our study. The observed significant PFS benefit and the reduction in the risk of death, which nearly reached the conventional levels of statistical significance, suggest that triplet therapy could potentially improve OS in mHSPC patients.

In the subgroup analysis stratified by the disease volume, triplet therapy was associated with a 36% reduction in the disease progression risk compared to ARPI-based doublet therapy in the patients with high-volume disease, while no statistically significant improvement was observed in the patients with low-volume disease. Based on the treatment rankings, Abi + Doc + ADT had the highest likelihood of improving PFS among the patients with high-volume disease, while Dar + Doc + ADT and Abi + Doc + ADT had the highest and second-highest likelihood of improving OS in the same setting. Conversely, Enz + ADT had the highest likelihood of improving both PFS and OS in low-volume disease. These updated results, which include the ARANOTE trial, are consistent with previous NMA findings [4]. High-volume disease is generally associated with biologically and clinically aggressive characteristics, including a higher probability of harboring androgen receptor-independent cells, suggesting a possible rationale for the efficacy of adding cytotoxic chemotherapy to ARPI + ADT in high-volume disease [31]. Supporting this concept, the PFS and OS benefit that resulted from adding Doc to ADT was most prominent in the patients with high-volume disease in the CHAARTED trial [18]. Overall, current evidence suggests that triplet therapy should be considered for patients with high-volume disease who can tolerate it.

It is essential to consider both effectiveness and potential AEs in the treatment decision-making process [32,33]. Our NMA revealed that triplet therapy did not significantly increase the risk of any-grade or grade ≥ 3 AEs compared to doublet therapies. Based on our treatment ranking, Abi + Doc + ADT had the lowest likelihood of increasing the risk of grade ≥ 3 AEs. Due to its low blood–brain barrier penetration and limited potential for drug–drug interactions, the safety profile of Dar was confirmed in the ARAMIS [34] and ARASENS [16] trials. In our analysis, Dar + ADT demonstrated a favorable safety profile regarding grade ≥ 3 AEs and fatigue, a common AE of ARPIs. Hematologic AEs, such as anemia, neutropenia, and febrile neutropenia (FN), were more common in Doc-based doublet therapy and triplet therapies. However, it is important to note that the patients enrolled in RCTs were generally younger and fitter than those seen in real-world clinical practice, which may lead to an underestimation of the toxicity burden. The real-world data indicate that neutropenia and FN are the primary reasons for treatment discontinuation or dose reduction [35]. Therefore, managing hematologic AEs is important for patients treated with triplet therapy.

Several limitations in our study should be noted. First, our analyses included studies with various patient populations, which could introduce potential heterogeneity into the results. Second, we included only studies investigating combinations of ARPI, Doc, and ADT, while studies assessing new agents such as ^177^Lu-PSMA and Rucaparib [36,37] are emerging. These studies were not included in the current analysis as they are still phase Ⅱ RCTs. However, when a phase Ⅲ RCT becomes available, future evaluations should consider including these regimens alongside ARPI, Doc, and ADT. Third, the follow-up duration varied across the included studies, potentially affecting the number of survival events. Most notably, the ARANOTE trial [6], which is the most recent RCT, had a short follow-up period, which may have been insufficient to adequately evaluate PFS. Additionally, while we discussed the potential of PFS as a surrogate marker for OS, it is important to note that the current evidence does not include triplet therapy. This highlights the need for the cautious interpretation of PFS as a surrogate marker for OS in this context. Fourth, the impact of triplet therapy on oncological outcomes in patients with low-volume disease remains unclear. In most of the studies, the number of patients with low-volume disease was less than 40% of the cohort, leading to a low statistical power in our analysis. Fifth, while we investigated PFS and OS stratified by the disease volume, other factors such as synchronous vs. metachronous, Gleason pattern 5, and TP53 mutations also influence mortality, even for patients with low-volume disease [38,39,40]. Therefore, patients with these factors may benefit from triplet therapy, even in low-volume disease contexts. Finally, in the analysis of PFS and OS, we extracted data of prior/absent Doc from the ARCHES [17], ENZAMET [19], and TITAN [26] trials; however, this was not feasible in the AE analysis due to the lack of stratified data. In the ARCHES [17] and TITAN [26] trials, Doc was administered before ARPI initiation, likely minimizing its impact on AE incidence. Conversely, in the ENZAMET trial [19], Doc was administered concomitantly with Enz + ADT. Additionally, the ENZAMET trial [19] included non-steroidal anti-androgen therapy with ADT in the control arm, potentially providing a differential survival benefit in the control arm and weighing against both the survival and the AE outcomes of Enz.

## 5. Conclusions

By synthesizing the most recently updated data, we confirmed that triplet therapies can improve the clinical outcomes of patients with mHSPC compared with the currently available ARPI- and Doc-based doublet treatment regimens without significantly increasing the rates of severe AEs.

There is robust evidence confirming the efficacy of triplet therapies in patients with high-volume disease, while its benefit in patients with low-volume disease remains uncertain to date. These findings warrant further confirmation in head-to-head trials with an extended follow-up powered for OS, especially in patients with low-volume disease.

## Figures and Tables

**Figure 1 cancers-17-00205-f001:**
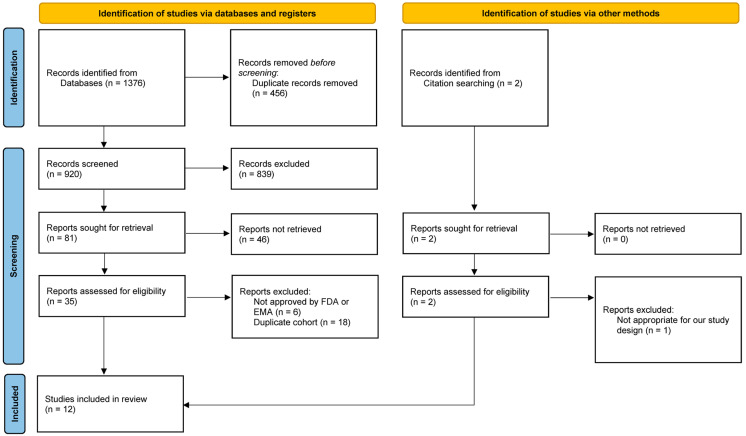
PRISMA 2020 flow diagram.

**Figure 2 cancers-17-00205-f002:**
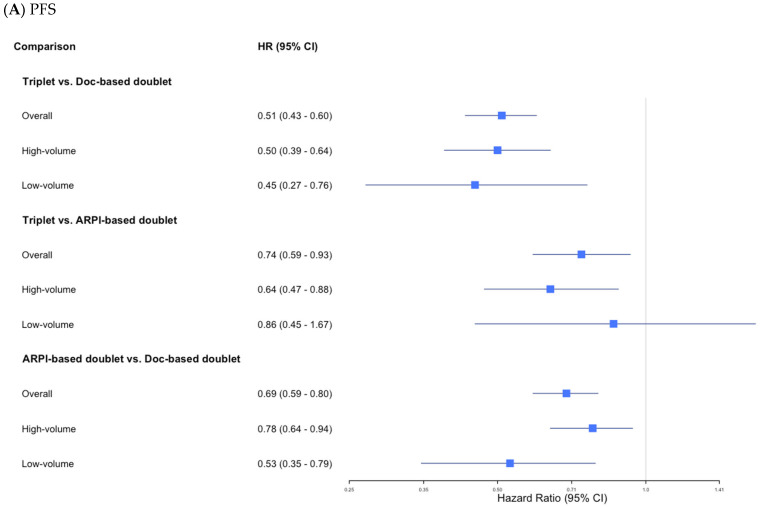
Network meta-analyses, (**A**) PFS, (**B**) OS, and (**C**) AEs. Abbreviations: Abi = abiraterone; ADT = androgen deprivation therapy; AE = adverse event; Apa = apalutamide; ARPI = androgen receptor pathway inhibitor; CI = confidence interval; Dar = darolutamide; Doc = docetaxel; Enz = enzalutamide; HR = hazard ratio; OS = overall survival; PFS = progression-free survival; RR = risk ratio.

**Figure 3 cancers-17-00205-f003:**
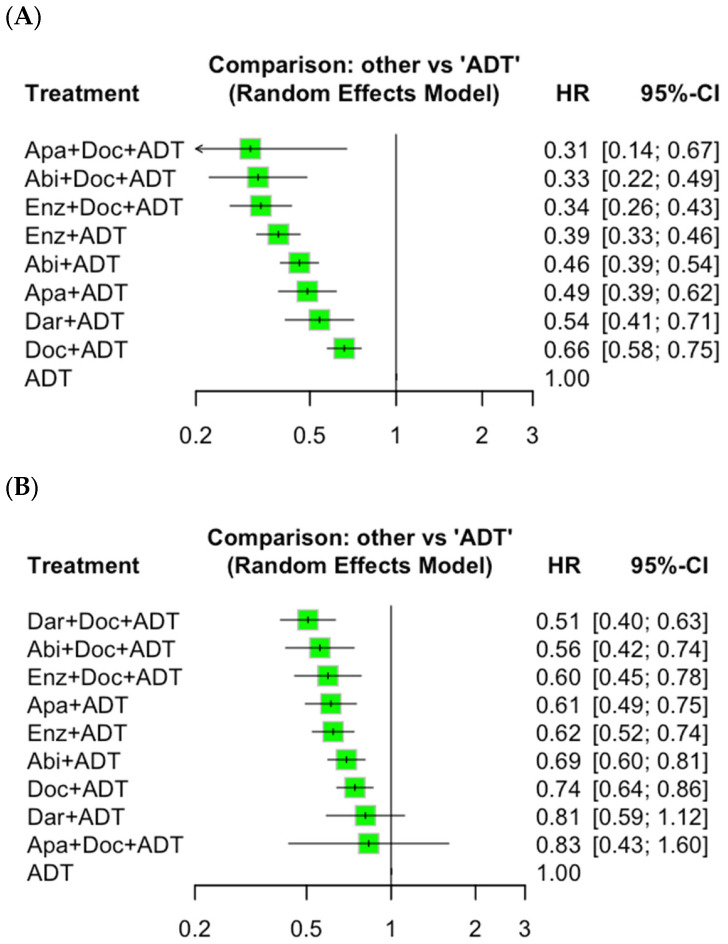
Forest plot of network meta-analyses of individual regimens, (**A**) PFS and (**B**) OS. Abbreviations: Abi = abiraterone; ADT = androgen deprivation therapy; Apa = apalutamide; Dar = darolutamide; Doc = docetaxel; Enz = enzalutamide; OS = overall survival; PFS = progression-free survival.

**Table 1 cancers-17-00205-t001:** Study demographics of included studies.

Study	ARANOTE	ARASENS	ARCHES		CHAARTED	ENZAMET		GETUG-AFU15	LATITUDE	PEACE-1	STAMPEDE (Arm B,C,E)	STAMPEDE (Arm C,G)	STAMPEDE (Arm G)	TITAN	
Author	Saad et al. [6]	Hussain et al. [16]	Armstrong et al. [17]		Kyriakopoulos et al. [18]	Sweeney et al. [19]		Gravis et al. [20]	Fizazi et al. [21]	Fizazi et al. [22]	Clarke et al. [23]	Sydes et al. [24]	James et al. [25]/Hoyle et al. [28]	Chi et al. [26]	
Year	2024	2023	2022		2018	2023		2016	2019	2022	2019	2018	2017/2019	2021	
Treatment Arm	Darolutamide + ADT	Darolutamide + Docetaxel + ADT	Enzalutamide + ADT	Enzalutamide + Docetaxel + ADT	Docetaxel + ADT	Enzalutamide + ADT	Enzalutamide + Docetaxel + ADT	Docetaxel + ADT	Abiraterone + ADT	Abiraterone + Docetaxel + ADT	ADT + Docetaxel	Abiraterone + ADT	Abiraterone + ADT	Apalutamide + ADT	Apalutamide + Docetaxel + ADT
Control Arm	Placebo + ADT	Placebo + ADT + Docetaxel	Placebo + ADT	Placebo + Docetaxel + ADT	ADT	NSAA + ADT	NSAA + Docetaxel + ADT	ADT	Placebo + ADT	Docetaxel + ADT	ADT	Docetaxel + ADT	ADT	Placebo + ADT	Placebo + Docetaxel + ADT
No. of Patients	669	1305	945	205	790	622	503	385	1199	710	1086	566	1917	939	113
Treatment	446	651	471	103	397	310	253	192	597	355	362	377	960	467	58
Control	223	654	474	102	393	312	250	193	602	355	724	189	957	472	55
Age															
Treatment	70 (43–93)	67 (42–85)	70 (46–92)		64 (36–88)	69 (63–74)		63 (57–68)	67.3 ± 8.5	66 (37–85)	65 (60–70)	66 (62–71)	67 (42–85)	69 (45–94)	
Control	70 (45–91)	67 (41–89)	69 (51–89)		63 (39–91)	69 (64–75)		64 (58–70)	66.8 ± 8.7	66 (44–84)	65 (61–70)	66 (61–70)	67 (39–84)	68 (43–90)	
Disease Volume (High/Low, %)															
Treatment	70.6/29.4	76.3/23.7	62/38		66/34	53/47		48/52	82/18	63/37	54/46	NA	54/46	62/38	
Control	70.4/29.6	77.7/22.3	65/35		64/36	54/46		47/53	78/22	65/35	57/43	NA	51/49	65/35	
HR for PFS	rPFS	Time to CRPC	rPFS		cPFS	cPFS		rPFS	rPFS	rPFS	rPFS	rPFS	PFS	rPFS	
	0.54 (0.41–0.71)	0.36 (0.30–0.42)	0.37 (0.28–0.49)	0.52 (0.30–0.89)	0.62 (0.51–0.75)	0.40 (0.32–0.50)	0.51 (0.41–0.64)	0.69 (0.55–0-87)	0.47 (0.39–0.55)	0.54 (0.44–0.67)	0.69 (0.59–0.81)	0.65 (0.48–0.88)	0.45 (0.37–0.54)	0.49 (0.39–0.62)	0.47 (0.22–1.01)
HR for OS	0.81 (0.59–1.12)	0.686 (0.571–0.822)	0.64 (0.51–0.81)	0.74 (0.46–1.20)	0.72 (0.59–0.89)	0.60 (0.47–0.78)	0.82 (0.63–1.06)	0.88 (0.68–1.14)	0.66 (0.56–0.78)	0.75 (0.59–0.95)	0.81 (0.69–0.95)	1.16 (0.82–1.65)	0.61 (0.49–0.79)	0.61 (0.50–0.76)	1.12 (0.59–2.12)
Follow-up (months), median (Treatment/Control)	25.3/25.0	43 (43.7/42.4)	44.6		53.7	68		83.9	51.8	45.7 (46.2/45.0)	78.2	48	40	44	

Abbreviations: ADT = androgen deprivation therapy; cPFS = clinical progression-free survival; CRPC = castration-resistant prostate cancer; HR = hazard ratio; NA = not available; NSAA = non-steroidal anti-androgen; OS = overall survival; PFS = progression-free survival; rPFS = radiographic progression-free survival.

**Table 2 cancers-17-00205-t002:** Summary of network meta-analyses on PFS and OS.

	Abi + Doc + ADT			Apa + Doc + ADT			Dar + Doc + ADT			Enz + Doc + ADT		
Comparator HR (95% CI)	Overall	High Volume	Low Volume	Overall	High Volume	Low Volume	Overall	High Volume	Low Volume	Overall	High Volume	Low Volume
**PFS**												
Abi + ADT	0.72 (0.48–1.08)	0.62 (0.38–1.00)	0.90 (0.39–2.04)	0.67 (0.31–1.48)	NA	NA	NA	NA	NA	0.73 (0.56–0.97)	0.67 (0.47–0.96)	0.57 (0.27–1.21)
Apa + ADT	0.67 (0.43–1.06)	NA	NA	0.63 (0.28–1.42)						0.69 (0.49–0.96)	NA	NA
Dar + ADT	0.61 (0.38–0.99)	0.47 (0.27–0.82)	1.44 (0.50–4.14)	0.57 (0.25–1.30)						0.62 (0.43–0.90)	0.51 (0.33–0.80)	0.92 (0.34–2.51)
Enz + ADT	0.85 (0.55–1.30)	0.75 (0.42–1.34)	1.55 (0.63–3.81)	0.80 (0.36–1.76)						0.87 (0.64–1.17)	0.81 (0.50–1.32)	0.99 (0.43–2.28)
Doc + ADT	0.50 (0.35–0.72)	0.47 (0.30–0.73)	0.58 (0.28–1.20)	0.47 (0.22–1.01)						0.51 (0.42–0.63)	0.51 (0.38–0.69)	0.37 (0.19–0.70)
ADT	0.33 (0.22–0.49)	0.28 (0.18–0.45)	0.43 (0.20–0.93)	0.31 (0.14–0.67)						0.34 (0.26–0.43)	0.31 (0.22–0.43)	0.28 (0.14–0.55)
**OS**												
Abi + ADT	0.80 (0.59–1.09)	0.85 (0.60–1.20)	1.11 (0.59–2.09)	1.20 (0.62–2.34)	NA	NA	0.73 (0.56–0.94)	0.82 (0.62–1.07)	0.91 (0.49–1.71)	0.86 (0.64–1.16)	1.03 (0.72–1.46)	0.82 (0.40–1.66)
Apa + ADT	0.91 (0.64–1.30)	NA	NA	1.36 (0.68–2.72)			0.83 (0.61–1.13)	NA	NA	0.98 (0.69–1.38)	NA	NA
Dar + ADT	0.69 (0,45–1.05)	0.65 (0.41–1.04)	0.84 (0.30–2.34)	1.03 (0.49–2.13)			0.62 (0.42–0.92)	0.63 (0.41–0.95)	0.69 (0.25–1.91)	0.73 (0.48–1.12)	0.79 (0.49–1.26)	0.62 (0.21–1.80)
Enz + ADT	0.90 (0.65–1.25)	0.76 (0.48–1.20)	1.48 (0.75–2.90)	1.34 (0.68–2.64)			0.81 (0.61–1.08)	0.73 (0.48–1.10)	1.21 (0.62–2.37)	0.96 (0.69–1.32)	0.91 (0.57–1.46)	1.09 (0.52–2.29)
Doc + ADT	0.75 (0.59–0.95)	0.72 (0.55–0.95)	0.83 (0.50–1.38)	1.12 (0.59–2.12)			0.68 (0.57–0.81)	0.69 (0.58–0.83)	0.68 (0.41–1.13)	0.80 (0.64–1.01)	0.87 (0.65–1.16)	0.61 (0.33–1.11)
ADT	0.56 (0.42–0.74)	0.52 (0.38–0.71)	0.75 (0.43–1.31)	0.83 (0.43–1.60)			0.51 (0.40–0.63)	0.50 (0.40–0.63)	0.62 (0.36–1.07)	0.60 (0.45–0,78)	0.63 (0.46–0.87)	0.55 (0.29–1.05)

Abbreviations: Abi = abiraterone; ADT = androgen deprivation therapy; Apa = apalutamide; CI = confidence interval; Dar = darolutamide; Doc = docetaxel; Enz = enzalutamide; HR = hazard ratio; NA = not available; OS = overall survival; PFS = progression-free survival.

**Table 3 cancers-17-00205-t003:** Summary of treatment ranking.

	Summary of Included Studies	Treatment Ranking (P-Score)
**PFS**		
Overall	9 studies 12 comparisons 7646 patients	Enz + Doc + ADT (0.833) > Abi + Doc + ADT (0.829) > Apa + Doc + ADT (0.80) > Enz + ADT (0.69) >Abi + ADT (0.48) > Apa + ADT (0.41) > Dar + ADT (0.31) > Doc + ADT (0.14) > ADT (<0.001)
High volume	8 studies 9 comparisons 7881 patients	Abi + Doc + ADT (0.90) > Enz + Doc + ADT (0.86) > Enz + ADT (0.69) > Abi + ADT (0.52) > Dar + ADT (0.27) > Doc + ADT (0.25) > ADT (<0.001)
Low volume		Enz + ADT (0.81) > Enz + Doc + ADT (0.80) > Dar + ADT (0.75) > Abi + Doc + ADT (0.52) > Abi + ADT (0.43) > Doc + ADT (0.18) > ADT (<0.01)
**OS**		
Overall	10 studies 13 comparisons 8951 patients	Dar + Doc + ADT (0.93) > Abi + Doc + ADT (0.79) > Enz + Doc + ADT (0.69) > Apa + ADT (0.66) > Enz + ADT (0.63) > Abi + ADT (0.43) > Doc + ADT (0.30) > Apa + Doc + ADT (0.29) > Dar + ADT (0.25) > ADT (0.04)
High volume	9 studies 10 comparisons 9504 patients	Dar + Doc + ADT (0.91) > Abi + Doc + ADT (0.84) > Abi + ADT (0.63) > Enz + Doc + ADT (0.57) > Enz + ADT (0.45) > Doc + ADT (0.33) > Dar + ADT (0.25) > ADT (0.02)
Low volume		Enz + ADT (0.85) > Enz + Doc + ADT (0.75) > Dar + Doc + ADT (0.67) > Abi + ADT (0.58) > Abi + Doc + ADT (0.46) > Dar + ADT (0.33) > Doc + ADT (0.25) > ADT (0.12)

Abbreviations: Abi = abiraterone; ADT = androgen deprivation therapy; Apa = apalutamide; Dar = darolutamide; Doc = docetaxel; Enz = enzalutamide; OS = overall survival; PFS = progression-free survival.

## Data Availability

The data presented in this study are available in this article and Appendix A.

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
