# Peer review of "An Updated Systematic Review and Network Meta-Analysis of First-Line Triplet vs. Doublet Therapies for Metastatic Hormone-Sensitive Prostate Cancer"

_cancers, 2025, doi:10.3390/cancers17020205_

Round 1
Reviewer 1 Report
Comments and Suggestions for Authors
The authors have performed an excellent and up to date synthesis of the data exploring triplet combinations for the management of prostate cancer. The scientific methodology is sound and the results are presented in a clear and concise way. Overall, the analysis has significant merit but would benefit from adding some nuance to the discussion.
In general, there are two major questions regarding the management of metastatic hormone sensitive prostate cancer.
The first pertains to whether there truly is an overall survival benefit to triplet regimens compared to doublets of GnRH agonists/antagonists and novel androgen receptor signaling inhibitors. A trial directly exploring the latter has yet to be conducted. The authors have chosen PFS as their main outcome.
The second is whether there is a readily identifiable subset of patients with a good response to hormonal therapy alone and which could forego chemotherapy.
The current analysis attempts to answer both these questions in part, but does not add much novelty to the debate due to the study adding limited novel information compared to the authors' prior analyses on this topic (and especially noting that the follow up time is short even for the ARANOTE trial which is the main addition to this analysis) and due to the lack of granularity in data obtained from the trials. It's also notable, and should be better highlighted in the discussion, that the ARASENS trial was not included in the analysis due to incomplete data. The ultimate debate is whether introduction of chemotherapy at the time of progression yields similar OS as its incorporation upfront. A clear OS benefit is not obvious from this data, although the authors point out that the comparisons are approaching statistical significance. This is likely a sign of heterogenous populations, some who benefit and some who do not from chemotherapy, and an attempt must be made to identify these patients. A blanket statement, especially with statistical tests not crossing signifiance threshold, that triplets improve survival over modern doublets should be made with great caution. Additionally, the authors might want to further stress caution about the choice of PFS based on its potential surrogacy for OS. In light of already negative statistical comparisons, it should be noted in the discussion that 1) the surrogacy analysis did not include triplet regimens, and 2) that the surrogacy endpoints have their own uncertainty built in, which compounds with that shown by the authors' own analysis. The comparison in AEs is important for this debate, as the main concern with triplets vs doublets is the toxicity burden. While this analysis did not show significant differences in different grades of adverse effects which is encouraging, it is notable that these are trial patients who are generally younger and fitter than the real world population and clinical experience speaks to the poorer tolerability of taxane based treatments with their afferent risks of neuropathy, greater fatigue, and (febrile) neutropenia.
The second question of patient selection is addressed solely by dichotomizing the population into high- or low-volume disease using CHAARTED criteria. The issue is that this (albeit standard) approach leaves out important variables such as PSA response by 6-7 months, de novo vs metachronous metastasis etc. which have been consistently shown to predict outcomes more strongly than CHAARTED criteria (for some). The authors could attempt a subgroup analysis of trials which did provide this data such as CHAARTED, ARASENS or TITAN to increase the novelty factor of this particular re-analysis. In any case, this should be emphasized to a greater degree in a more nuanced discussion.
Author Response
Reviewer #1:
The authors have performed an excellent and up to date synthesis of the data exploring triplet combinations for the management of prostate cancer. The scientific methodology is sound and the results are presented in a clear and concise way. Overall, the analysis has significant merit but would benefit from adding some nuance to the discussion.
Author Response: We appreciate the reviewer’s thoughtful comments which have substantially improved the content and clarity of this piece. Please find below our detailed response to the reviewer’s comments:
- In general, there are two major questions regarding the management of metastatic hormone sensitive prostate cancer. The first pertains to whether there truly is an overall survival benefit to triplet regimens compared to doublets of GnRH agonists/antagonists and novel androgen receptor signaling inhibitors. A trial directly exploring the latter has yet to be conducted. The authors have chosen PFS as their main outcome. The second is whether there is a readily identifiable subset of patients with a good response to hormonal therapy alone and which could forego chemotherapy. The current analysis attempts to answer both these questions in part, but does not add much novelty to the debate due to the study adding limited novel information compared to the authors' prior analyses on this topic (and especially noting that the follow up time is short even for the ARANOTE trial which is the main addition to this analysis) and due to the lack of granularity in data obtained from the trials. It's also notable, and should be better highlighted in the discussion, that the ARASENS trial was not included in the analysis due to incomplete data.
Author Response: Thank you very much for your insightful comment. As you pointed out, the novelty of our current analyses may be limited compared to our previous analyses. However, to our knowledge, this is the first analysis that incorporates the data of Dar + ADT and provides treatment rankings. We believe this adds value by offering a comprehensive perspective on treatment efficacy, despite the aforementioned limitations. Additionally, we have addressed in the Evidence Synthesis section that ARASENS trial was excluded due to incomplete data (Line 146-148).
- The ultimate debate is whether introduction of chemotherapy at the time of progression yields similar OS as its incorporation upfront.
Author Response: We appreciate your valuable comment and agree that this is a critical question in the management of mHSPC. However, due to the lack of strong evidence comparing the introduction of chemotherapy at progression vs. upfront incorporation, we have not discussed this point. We acknowledge this as an important area for future research.
- A clear OS benefit is not obvious from this data, although the authors point out that the comparisons are approaching statistical significance. This is likely a sign of heterogenous populations, some who benefit and some who do not from chemotherapy, and an attempt must be made to identify these patients. A blanket statement, especially with statistical tests not crossing signifiance threshold, that triplets improve survival over modern doublets should be made with great caution. Additionally, the authors might want to further stress caution about the choice of PFS based on its potential surrogacy for OS. In light of already negative statistical comparisons, it should be noted in the discussion that 1) the surrogacy analysis did not include triplet regimens, and 2) that the surrogacy endpoints have their own uncertainty built in, which compounds with that shown by the authors' own analysis.
Author Response: Thank you very much for your insightful comments. As you pointed out, careful interpretations of PFS results is essential. We have rephrased the limitation (Line 336-339) as follows:
“Third, follow-up duration varied across the included studies, potentially affecting the number of survival events. Especially, the ARANOTE trial [1], which is the most recent RCT, had a short follow-up period, which may have been insufficient to adequately evaluate even PFS. Additionally, while we discussed the potential of PFS as a surrogate marker for OS, it is important to note that the current evidence does not include triplet therapy. This highlights the need for cautious interpretation of PFS as a surrogate marker for OS in this context.”
- The comparison in AEs is important for this debate, as the main concern with triplets vs doublets is the toxicity burden. While this analysis did not show significant differences in different grades of adverse effects which is encouraging, it is notable that these are trial patients who are generally younger and fitter than the real world population and clinical experience speaks to the poorer tolerability of taxane based treatments with their afferent risks of neuropathy, greater fatigue, and (febrile) neutropenia.
Author Response: We appreciate your valuable comment. To address the difference of real-world data, we have revised as follows (Line):
“However, it is important to note that patients enrolled in RCTs are generally younger and fitter than those seen in real-world clinical practice, which may lead to an underestimation of the toxicity burden.”
The second question of patient selection is addressed solely by dichotomizing the population into high- or low-volume disease using CHAARTED criteria. The issue is that this (albeit standard) approach leaves out important variables such as PSA response by 6-7 months, de novo vs metachronous metastasis etc. which have been consistently shown to predict outcomes more strongly than CHAARTED criteria (for some). The authors could attempt a subgroup analysis of trials which did provide this data such as CHAARTED, ARASENS or TITAN to increase the novelty factor of this particular re-analysis. In any case, this should be emphasized to a greater degree in a more nuanced discussion.
Author Response: Thank you very much for your insightful comment. Unfortunately, the ARANOTE trial did not provide separate data for de novo and metachronous cases. As such, we were unable to conduct a subgroup analysis focusing on these categories. Excluding the ARANOTE trial would have resulted in the same analysis as our previous study. Additionally, we have emphasized the importance of other prognostic factors in the limitations section (Line 345-348).
Reviewer 2 Report
Comments and Suggestions for Authors
In addition to the authors' previous article, the PFS and OS results of Triplet and Doublet therapy were reported separately for low and high volume, adding the ARANOTE trial results. The article covers past clinical trials, is clearly presented, and there are no corrections that need to be pointed out.
Author Response
Reviewer #2:
In addition to the authors' previous article, the PFS and OS results of Triplet and Doublet therapy were reported separately for low and high volume, adding the ARANOTE trial results. The article covers past clinical trials, is clearly presented, and there are no corrections that need to be pointed out.
Author Response: Thank you very much for your kind words and positive feedback. We are glad that you found our manuscript clear and comprehensive. Your encouraging comments are greatly appreciated.
Reviewer 3 Report
Comments and Suggestions for Authors
The topic of the manuscript is interesting, and tries to shed light on a topic that is currently much debated; the authors present the results in an exhaustive way, so I do not find specific problems along the manuscript, which deserves to be published in the present form.
Author Response
Reviewer #3:
The topic of the manuscript is interesting, and tries to shed light on a topic that is currently much debated; the authors present the results in an exhaustive way, so I do not find specific problems along the manuscript, which deserves to be published in the present form.
Author Response: Thank you very much for your encouraging comments. We are delighted that you found our manuscript both interesting and comprehensive. Your support is greatly appreciated.
Round 2
Reviewer 1 Report
Comments and Suggestions for Authors
The authors have addressed all concerns and have strengthened the manuscript.